# Identifying Eigen-like hydrated protons at negatively charged interfaces

Eric Tyrode [ID] [1]*, Sanghamitra Sengupta [ID] [1] & Adrien Sthoer [ID] [1]

Despite the importance of the hydrogen ion in a wide range of biological, chemical, and physical processes, its molecular structure in solution remains lively debated. Progress has been primarily hampered by the extreme diffuse nature of the vibrational signatures of hydrated protons in bulk solution. Using the inherently surface-specific vibrational sum frequency spectroscopy technique, we show that at selected negatively charged interfaces, a resolved spectral feature directly linked to the $H_3O^+$ core in an Eigen-like species can be readily identified in a biologically compatible pH range. Centered at ~2540 cm$^{-1}$, the band is seen to shift to ~1875 cm$^{-1}$ when forming $D_3O^+$ upon isotopic substitution. The results offer the possibility of tracking and understanding from a molecular perspective the behavior of hydrated protons at charged interfaces.

---

[1] Department of Chemistry, KTH Royal Institute of Technology, SE-10044 Stockholm, Sweden. *email: tyrode@kth.se

T he hydrated proton (H$^+$) in concert with the OH$^-$, is arguably the most important ion in aqueous solution, playing a prominent, if not, vital role in a variety of biological, chemical, and geological processes. It represents the essence of a concept as widely used and popularized as pH, and is obviously fundamentally involved in all acid/base chemistry. Despite recent experimental[1–7] and theoretical advances[4,8–10], a comprehensive molecular understanding of hydrated protons in solution remains elusive. Vibrational spectroscopy, being sensitive to hydrogen bond strength and dynamics, has been the method of choice for determining the molecular structure of the hydrated proton[10]. However, concentrated acid solutions give rise to extremely diffuse vibrational signatures that extend over at least 2000 cm$^{-1}$ [4,10], a fact that reflects the wide-ranging molecular distortions expected in the structural motifs, but at the same time makes the bands challenging to assign to specific species or conformers. In this sense, the structure of the hydrated proton in terms of Eigen[11] (H$_9$O$_4^+$), Zundel[12,13] (H$_5$O$_2^+$) or more delocalized motifs continues to be lively debated[10]. An important improvement in our understanding came from gas-phase studies of clusters of hydrated protons H$^+$(H$_2$O)$_n$ cryogenically cooled to 20 K[1,2,14]. The clusters limit the motifs accessible to the proton, generating well-resolved IR spectral features that have been directly linked to Zundel and Eigen structures, and are also used as a benchmark for testing the accuracy of different computational methods[8,10,15]. Though useful, the results from low-temperature gas-phase clusters are not necessarily transposable to condensed phases at ambient conditions[6,16]. At room temperature, we require background subtraction and the deconvolution of vibrational spectral features to obtain indirect information of the structural motifs[4,15,16]. A precious insight has been provided by ultrafast two-dimensional infrared (2D-IR) spectroscopy, where by connecting the coupled resonances at different frequencies, distinct species have been identified in solution[6,7]. The current consensus leans towards the proton in the bulk liquid being primarily hydrated by two flanking water molecules in a Zundel-like configuration[4,7].

The scenario can be different at surfaces. The orientational constraints and asymmetric environment imposed by the interface may reduce the number of structural possibilities, and consequently, result in more resolved features. Vibrational sum frequency spectroscopy (VSFS), a second-order non-linear optical technique with an intrinsic surface specificity, offers the opportunity to yield vibrational spectra from molecules at interfaces[17]. Under the electric-dipole approximation, VSFS only detects molecules with a net polar orientation, which are typically found in the noncentrosymmetric environment of an interface[18]. The air/liquid surface of concentrated acid solutions has been the subject of several VSFS studies[19–23], but they have primarily focused on the OH stretching region of interfacial water molecules (>3000 cm$^{-1}$). Although some of the spectral changes observed, particularly at ~3200 cm$^{-1}$, were linked to the hydrated proton[21], other interpretations associate them instead to an increased number of probed water molecules in the diffuse double layer, which extends upon charging of the surface[22,24]. With the exception of a couple of studies on neat acid solutions[20,25], where no resolved resonant features were detected, the proton continuum region (<3000 cm$^{-1}$) on surfaces has largely been neglected.

Here, we use VSFS to probe the proton continuum spectral region of negatively charged liquid/vapor interfaces and identify resolved features associated with hydrated protons having a preferred orientation. The charged surfaces consist of Langmuir monolayers exposing either carboxylic acid or sulfate groups to solution. The associated negative surface potential results in surface hydronium ion concentrations that are orders of magnitude higher than in the bulk, making the spectral features

detectable at ambient pH. From the analysis of the VSF spectra at different polarization combinations and comparison with gas-phase cluster, 2D-IR, and Raman/IR MCR studies, we assign the band observed at ~2540 cm$^{-1}$ to the antisymmetric stretch of an Eigen-like species.

## Results

**Carboxylic acid Langmuir monolayers.** The starting point is the conventional homodyne SF spectra at the liquid/vapor interface of an arachidic acid Langmuir monolayer on a sodium chloride aqueous subphase at ambient pH (Fig. 1). The twenty carbon long fatty acid forms an insoluble monolayer that exposes the carboxylic acid headgroups to solution, and the hydrophobic alkyl chains towards the gas phase. The packing density can be varied by moving the barriers of the Langmuir trough. The spectra presented have been collected at a constant surface pressure of 20 mN m$^{-1}$, where the monolayer is found well-packed in a tilted condensed phase with an average area per molecule of 20 Å$^2$, as can be deduced from the surface pressure vs. molecular area isotherm (see Supplementary Note 1 and Supplementary Fig. 1). Although the intrinsic pK$_a$ of these fatty acids is similar to that of shorter chain carboxylic acids (i.e., pKa ~5), at the surface, when only the pH determining ions are present in solution, the apparent pK$_a$ is ~10.8[26]. The negative charge and accompanying surface potential result in surface proton concentrations that are several orders of magnitude higher than in the bulk. Moreover, the charging behavior upon addition of monovalent ion salts to solution has been shown to be consistent with the Gouy–Chapman theory for concentrations ≤50 mM[26,27].

Several molecular structural details of the interfacial region can be extracted from the spectra collected at the two different polarization combinations (i.e., SPS and SSP) shown in Fig. 1. The sharp peaks found between 2050 cm$^{-1}$ and 2250 cm$^{-1}$ are all linked to the alkyl chain of the fatty acid, and are primarily stretching modes of the terminal methyl group[26]. The absence of features from CD$_2$ groups, indicate that the monolayer is conformationally well-ordered in an all-*trans* configuration[28], consistent with the high packing densities of the tilted condensed phase. At lower frequencies, the peak at ~1720 cm$^{-1}$ observed in both polarization combinations, is assigned to the C=O stretch of the uncharged form of the carboxylic acid headgroup[26,29]. The spectra show no features linked to the charged carboxylate moieties, which are expected at ~1408 cm$^{-1}$ and ~1535 cm$^{-1}$ [26]. At pH ~6 and a NaCl concentration of 1 μM, the carboxylate modes are below the detection limit, yet the monolayer is expected to be ~0.5% deprotonated, as estimated from the Gouy–Chapman model[26]. Nevertheless, despite the relatively small surface charge (i.e., ~−4 mC m$^{-2}$), the low ionic strength of the subphase ensure a substantial surface potential that is in the order of ~−200 mV[26]. At higher frequencies, between 3100 cm$^{-1}$ and 3650 cm$^{-1}$, the OH stretching bands from water in the interfacial region are apparent (the OH stretch from the hydrogen bonded carboxylic acid gives rise to a weak broad band centered at ~2900–2950 cm$^{-1}$ [30]). The signal primarily originates from water molecules directly interacting with the carboxylic acid headgroup[26,31], with only a small contribution, if any, from water molecules in the diffuse double layer. At 1 μM the Debye length (i.e., ~300 nm) is significantly longer than the non-linear coherence length of the experimental geometry (Δk$_z^{-1}$ ≈ 50 nm), and most of the signal generated within the diffuse double layer cancels due to destructive interference[32,33].

**Spectral features of hydrated protons at charged interfaces.** Central to this study is the band observed at ~2540 cm$^{-1}$ in the SPS spectrum (Fig. 1b), which is located far from any vibrational

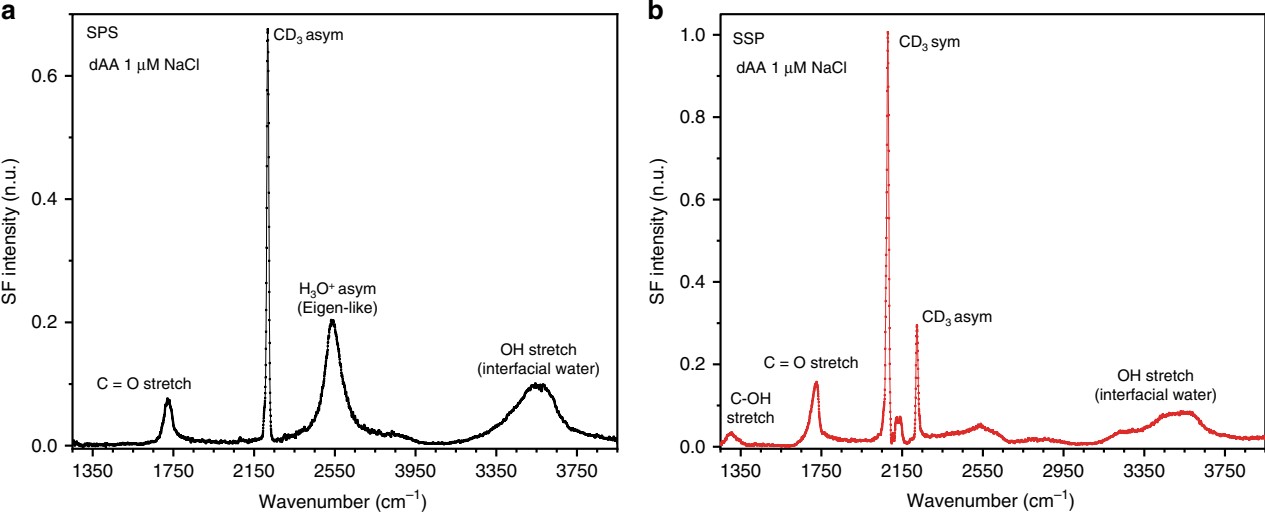

**Fig. 1 Interfacial spectral features of the carboxylic acid monolayer.** VSF spectra of a perdeuterated arachidic acid (dAA) Langmuir monolayer on a 1 μM NaCl solution at pH~6 collected under the (**a**) SPS, and (**b**) SSP polarization combinations. The different polarizations provide complementary information about the surface species and their orientation. The assignments are taken from refs. [26-29] and this study (see text). Measurements are carried at 22 °C and a constant surface pressure of 20 mN m$^{-1}$, corresponding to an area per molecule of ~20 Å$^2$.

mode expected from the fatty acid or aqueous subphase, but within the proton continuum range of concentrated acid solutions[10,20,25]. To confirm that the observed vibrational mode involves the displacement of an OH bond, isotopic substitution experiments were performed using $D_2O$. The two times heavier deuterium atoms cause the vibrational frequencies to shift within the harmonic level towards lower energies by a factor of ~1.36. The SF spectrum of an equivalent fatty acid monolayer on a $D_2O$ solution collected under the SPS polarization combination is shown in Fig. 2. In the spectrum, the broad water band that was in the case of $H_2O$ originally centered at ~3600 cm$^{-1}$ is seen, as expected, to red-shift to ~2650 cm$^{-1}$ in $D_2O$. More interestingly, a peak resolved at ~1875 cm$^{-1}$ proves that the ~2540 cm$^{-1}$ band indeed involves an OH stretching vibration. Compared to the OH stretches of water, typically observed between 3000 cm$^{-1}$ and 3600 cm$^{-1}$, the significant red-shift of the ~2540 cm$^{-1}$ band, indicates that the OH bonds are more strongly hydrogen-bonded, as is usually the case of water molecules surrounding the proton[4,14].

The peak position lies within the range generally assigned to embedded Eigen-like structures (i.e., an $H_3O^+$ core, symmetrically solvated by three water molecules)[1,4,6,14]. However, this classification is controversial given the extremely diffuse vibrational signature of the hydrated proton in bulk solution[4,7]. Although the cryogenically cooled cluster behavior cannot be readily extrapolated to the bulk liquid at room temperature, their resolved features provide useful insight. The charged delocalized Eigen cation $H_3O^+(H_2O)_3$ shows a characteristic peak centered at ~2660 cm$^{-1}$, unambiguously assigned to the doubly degenerate antisymmetric OH stretch of the Eigen core[1,3,9,14,34]. This band is also observed in larger gas clusters displaying an Eigen-like motif where the hydronium core is predominantly symmetrically solvated (i.e., $H^+(H_2O)_{n=9-12}$), and remains detectable in even bigger clusters (i.e., $n = 24$), though contributions red-shifted by up to ~600 cm$^{-1}$ also become apparent in the spectra[1,35]. In contrast, when the proton is shared by two water molecules in a Zundel-like arrangement $(H_2O\text{---}H\text{---}OH_2)^+$, vibrations involving the core proton in the cluster are further red-shifted by more than 1000 cm$^{-1}$[10]. 

Additional experimental information regarding the assignment and also molecular orientation of the hydrated proton at the

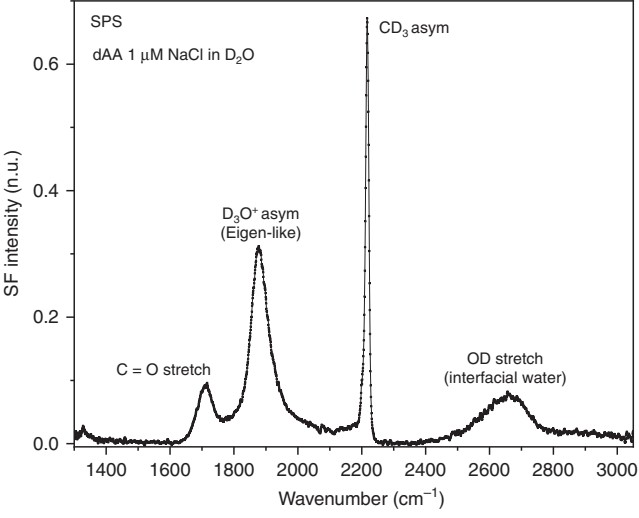

**Fig. 2 Isotopic substitution proves the OH stretching assignment.** VSF spectrum collected under the SPS polarization combination of a dAA monolayer on a 1 μM NaCl solution in $D_2O$ (pD ~6.4) at a constant surface pressure of 20 mN m$^{-1}$ (area per molecule ~20 Å$^2$). Temperature. 22 °C. The assignments are taken from[26,28] and this study (see text).

surface can be extracted from the SF spectra gathered at different polarization combinations[36,37]. The hydrated proton feature is strongly observed in the SPS spectrum, but besides a weak non-resonant background it is essentially absent in the PPP polarization combination (Fig. 3a). This is consistent with the band being assigned to an antisymmetric OH stretch of the $C_{3v}$ symmetry $H_3O^+$ in the Eigen core, with its $C_3$ axis aligned at, or close to the surface normal, as shown in the theoretical orientational analysis presented in Fig. 3b. The curves depict the expected variations in relative intensities as a function of the tilt angle for the polarization combinations SPS, SSP, and PPP, assuming a delta and a narrow Gaussian distribution of angles (see Supplementary Note 2 for details). However, though the experimental intensity in SSP is significantly weaker than for SPS (Fig. 3a), it is not negligible as predicted by the theoretical

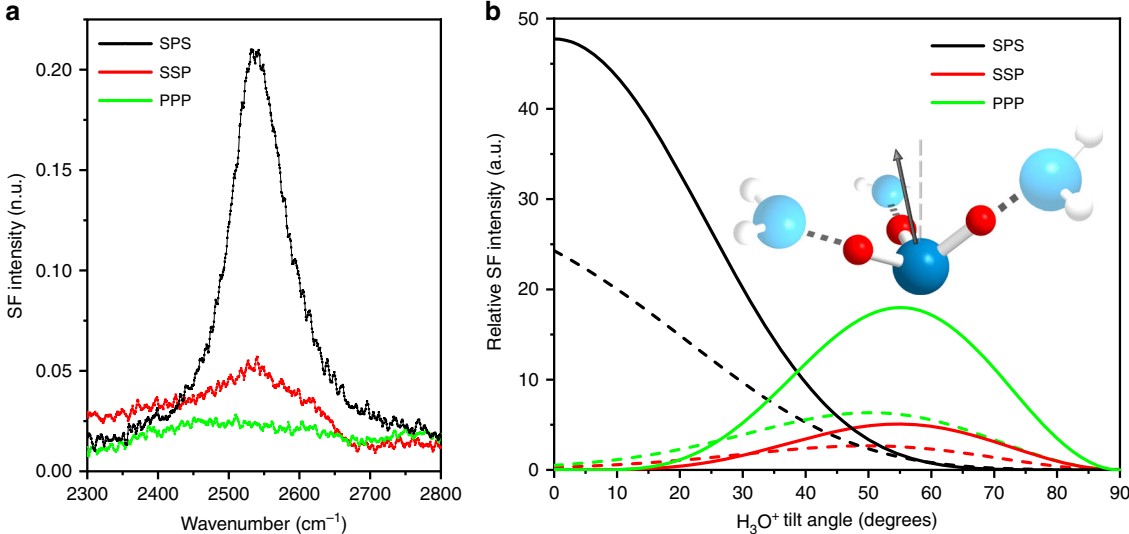

**Fig. 3 Polarization dependence and orientational analysis of the hydrated Eigen proton. a** VSF spectra collected in the polarization combinations SPS, SSP, and PPP of a dAA monolayer on a 1 μM NaCl solution in $H_2O$ (surface pressure 20 mN m$^{-1}$). **b** Theoretical curves showing the expected variations of the SF intensity of the antisymmetric stretch of $H_3O^+$ ($C_{3v}$ symmetry) as a function of tilt angle of the $C_3$ axis from the surface normal for the three polarization combinations, assuming a delta distribution (solid lines), and Gaussian distribution with standard deviation of 15° (dashed lines). See Supplementary Note 2 for details of the model. The sketch depicts the orientation of the hydronium core with the arrow along the $C_3$ axis. Experimental observations are consistent with the $C_3$ axis aligned closed to the surface normal. In the sketch, the oxygen and hydrogen atoms of the core Eigen proton are depicted as blue and red spheres, respectively. In the flanking water molecules, oxygen atoms are in light blue, while hydrogens are illustrated as white spheres.

analysis for low tilt angles (Fig. 3b). The discrepancy can be accommodated by considering potential contributions from the symmetric stretch of the $C_{3v}$ Eigen core, which when aligned close to the surface normal, should be preferentially observed in the SSP polarized spectra (see Supplementary Note 2 for details)[36,38]. Although the symmetric stretch has not been experimentally resolved in any of the previous IR cluster studies, it has been theoretically estimated to lie just ~38 cm$^{-1}$ red-shifted[3], overlapping[39], or even blue-shifted[1] relative to the antisymmetric mode, with a cross-section that is almost two orders of magnitude lower than for the degenerate antisymmetric stretch[9,40]. Given that the SF intensity has a square dependence on the IR and Raman transition moments[36,37], the significantly lower IR cross-section will have an obvious effect on the intensity of the symmetric stretching mode. Moreover, even though the mode is expected to be stronger in Raman, due to the non-Condon effect[4,41], the contribution of the IR component relative to that of Raman in the SF cross-section, becomes more important at the lower OH stretching frequencies of the Eigen core. Consequently, both elements (i.e., low IR cross-section, and non-Condon effect), indicate that the SF cross-section of the $H_3O^+$ symmetric stretch will be significantly lower than for the antisymmetric mode.

**Fatty sulfate monolayers**. To further extend the implications of the results, spectra were also collected at negatively charged surfaces that have as chargeable groups other functionalities than the carboxylic acid moiety. The SPS polarization SF spectrum of a twenty-carbon fatty sulfate Langmuir monolayer is presented in Fig. 4. At the pH and selected surface pressure of the experiment, the fatty sulfate monolayer is found in a liquid expanded phase. With an intrinsically lower pKa the fatty sulfate, even at pH 3.8, is substantially more deprotonated, but also shows a higher affinity for the other monovalent counterions (see Supplementary Note 1 for details). The presence of the hydrated proton peak at

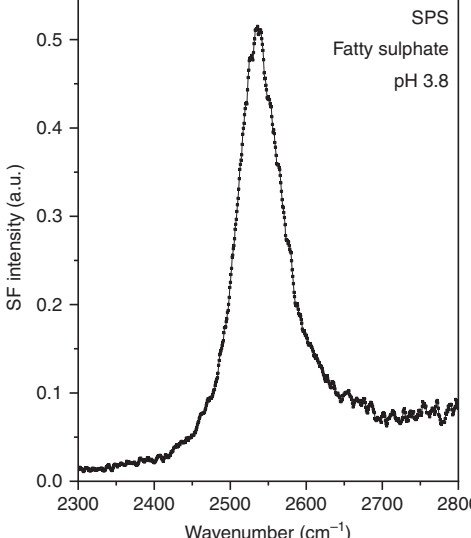

**Fig. 4 Hydrated protons at a negatively charged fatty sulfate interface.** VSF spectrum collected under the SPS polarization combination of a sodium eicosyl sulfate monolayer on a pH 3.8 HCl solution at a constant surface pressure of 2.5 mN m$^{-1}$ (area per molecule ~65 Å$^2$). Temperature 22 °C. The spectral feature assigned to the hydrated Eigen proton is observed in other negatively charged interfaces besides those involving carboxylic acid moieties.

~2540 cm$^{-1}$, proves that it is not unique to surfaces displaying charged carboxylic acid groups.

## Discussion
In contrast to bulk studies, the VSF results presented show compelling evidence of a narrow spectral feature linked to a specific hydrated proton configuration. The negative charge and

asymmetric environment imposed by the interface are key elements for this distinction. Under the electric-dipole approximation, VSFS only detects molecules with a net polar orientation that have vibrational modes that are simultaneously Raman and IR active[18,36]. This typically reduces the number of bands resolved when compared with the linear vibrational spectroscopies. Moreover, the surface electric field associated with the negatively charged interface plays a fundamental role, as it not only causes a substantial increase of the surface concentration of hydrated protons, but also has a direct influence on the adopted configuration and its average net orientation. For instance, the VSF spectrum of pure water, as well as that of solutions of HCl at pH 2, and HBr at pH −0.3, show no resolved resonant features in the proton continuum range[20,25], highlighting the importance of having a (partly) charged Langmuir monolayer for detecting the hydrated proton at the surface.

From the observed peak position and polarization dependence, together with the insight provided by the experimental and theoretical studies on small cold clusters, the ~2540 cm$^{-1}$ can be assigned to the degenerate antisymmetric OH stretch of the $H_3O^+$ core in an Eigen-like configuration, where the three OH are equivalent (i.e., $C_{3V}$ symmetry). The center position is also in agreement to that proposed from calculated IR and Raman spectra for structures that can be classified as Eigen-like[4]. The orientational analysis for the antisymmetric stretch presented in Fig. 3b, suggest that the principal axis of the $H_3O^+$ core should be oriented close to the surface normal. Although the net polar orientation can not be directly obtained from the homodyne SF spectra presented[42], the three hydrogens will most certainly be directed towards the negatively charged interface.

Interestingly, the Eigen-like VSF signal can be detected in fatty acid monolayers when the bulk hydronium ion concentration is just in the order of a few micromolar or less (i.e., pH ~6). At the negatively charged surface, however, the proton concentration is orders of magnitude higher. At low ionic strengths the charging behavior of the monolayer can be accurately predicted using the Gouy Chapman model[26], thus the surface concentration of $H_3O^+$ can be estimated to be ~2.5 mM in the presence of 1 μM NaCl in the subphase (see methods). At higher monovalent salt concentrations, despite an increase of the percentage of deprotonation of the fatty acid monolayer[26], the surface proton concentration is predicted to decrease due to a lowering of the surface potential as shown in Fig. 5a. In qualitative agreement with this prediction, VSF measurements carried out at higher ionic strengths show that the intensity of the $H_3O^+$ antisymmetric stretch indeed decreases with the concentration of NaCl in the subphase, at least up to 0.1 mM (Fig. 5b). The VSF intensity depends on the number of contributing oscillators squared, as well as their average orientation and distribution[36,43]. Provided the latter two remain constant, the VSF intensity is thus expected to be particularly sensitive to the surface concentration of Eigen-like protons (i.e., $N^2$ dependence). In fact, although the trends are systematically reproduced, significant variations in the absolute intensities of the hydrated proton band were observed in multiple repeat experiments. Ascribed to small variations in solution pH due to $CO_2$ absorption from the atmosphere, a detailed analysis of these fluctuations is clearly warranted but remain beyond the scope of this work.

The SF signal from the hydrated proton could, in principle, originate from Eigen-like species located in direct proximity to the interface, or further away within the diffuse double layer[32,44]. However, the fact that the Eigen band is more strongly observed on a 1 μM NaCl subphase than at higher salt concentrations (Fig. 5b), suggests that the origin is from close proximity to the interface. Had the signal been generated from within the diffuse double layer, due to destructive interference, it would have increased with ionic strength in the concentration range considered as experimentally confirmed elsewhere when probing water molecules in the diffuse layer[26,45] (i.e., the Debye screening length at 1 μM NaCl is significantly longer than the non-linear coherence length of the SF process[32,33]). Given that the Eigen-like proton is symmetrically hydrated, one possibility is that the hydrated proton forms a solvent separated ion pair[46] with the charged carboxylate and sulfate headgroups. Nonetheless, the hydrated proton must also interact with other competing counterions present at the interface. For the fatty acid monolayers, the proton has a significantly higher affinity than other monovalent cations such as $Na^+$ to interact with the carboxylate headgroup[47]. This is not the case for the sulfate headgroup[48], where the presence of sodium counterions in the subphase could have a detrimental effect on the measured $H_3O^+$ core signal.

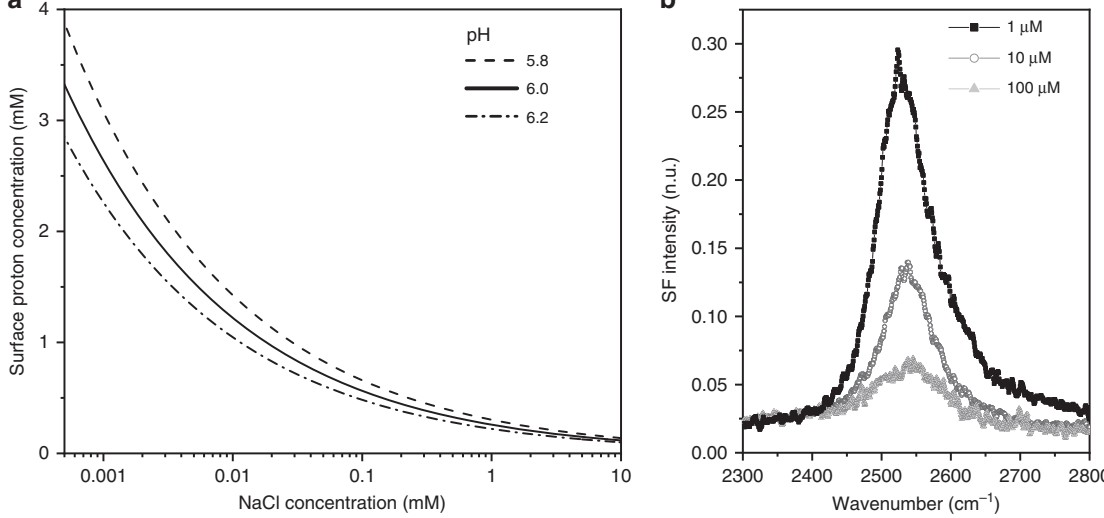

**Fig. 5 Salt dependence of the hydrated proton spectral feature: theory compared with experiments. a** Calculated surface proton concentration as a function of NaCl in the subphase of a fatty acid Langmuir monolayer based on the Gouy Chapman model, for three different pH. **b** VSF spectra in the SPS polarization of a dAA monolayer on an aqueous subphase with varying concentrations of NaCl at natural pH (surface pressure 20 mN m$^{-1}$, temperature 22 °C). Note that the $H_3O^+$ antisymmetric stretch intensity follows the same trend predicted for the surface proton concentration as a function of salt.

In summary, a resolved spectral feature assigned to the anti-symmetric OH stretch of the $H_3O^+$ core in an Eigen-like configuration has been identified at negatively charged interfaces exposing carboxylic acid and sulfate headgroups to solution. The fact that Eigen-like protons can be detected in a biologically relevant pH range, may offer novel perspectives for understanding essential physiological functions involving proton interactions with cell membrane interfaces[10]. The presence of a band linked to the charge carrier opens a window of opportunities for potentially tracking the behavior of hydrated protons in a variety of conditions and systems. Moreover, VSFS could also be used in its phase-sensitive[42], and time-resolved configurations[49,50], to extract additional information including absolute polar orientation and vibrational dynamics. We anticipate that Eigen-like protons are not necessarily the only structural motifs with a preferred orientation that can be found at an interface, but at given conditions, other configurations, in particular, Zundel-like hydrated protons could well be the preferred species. It is hoped that the information provided here may stimulate theoretical and simulation studies of hydrated protons at charged interfaces.

## Methods

**Materials and preparation of solutions**. NaCl (99.999% trace metal basis, Merck) was baked at 500 °C for 1 h in order to remove traces of organic contaminants. HCl 36.5% (99.999% trace metal basis, Alfa Aesar), deuterium oxide (D 99.9%, <0.8 μS cm$^{-1}$ in conductivity, Cambridge Isotope Laboratories, Inc., MA, USA), chloroform (anhydrous grade, stabilized with ethanol, Merck), methanol (anhydrous, 99.8%, Merck), eicosanoic-d39 acid (97%, dAA for deuterated arachidic acid, Merck), and sodium eicosyl sulfate (>95%, Santa Cruz Biotechnology, TX, USA) were used as received. The electrolyte solutions were prepared by serial dilution in either $D_2O$ or ultrapure water from an Integral 15 Millipore system (resistivity of 18.2 MΩ.cm and TOC < 3 ppb). The solvents used for the fatty acid and fatty sulfate spreading solutions consisted of pure chloroform, and a mixture of chloroform-methanol in a 10:1 proportion, respectively.

**Langmuir trough**. The monolayers were formed by spreading a few microliters of the fatty acid or fatty sulfate solutions on the surface of the subphase placed in a Langmuir trough (KSV NIMA, Finland, 195 mm length, 50 mm width, and 4 mm in depth). After a waiting time of at least 10 min to ensure full solvent evaporation, the monolayers were compressed at a rate of 5 mm per minute. The surface pressure was measured using a paper Wilhelmy plate. All the VSF experiments were carried out a constant surface pressure and temperature of 22.0 ± 0.5 °C. For the fatty acid case, the surface pressure was 20 mN m$^{-1}$ (~20 Å$^2$ per molecule), while for the fatty sulfate it was set to 2.5 mN m$^{-1}$ (~65 Å$^2$ per molecule). See Supplementary Note 1 for additional details.

**VSF spectrometer**. The femtosecond spectrometer has been described in detail elsewhere[51], and only a brief description of the key features is given here. The tuneable IR (~100 fs and 1 kHz) and visible pulses (805 nm, tuneable beam shaper, >20 ps long) are focused at the sample position in a co-propagating geometry at angles of incidence of 55° and 70°, and pulse energies of ~4 mJ cm$^{-2}$ and ~12 mJ cm$^{-2}$, respectively. The spectrometer features a high degree of automation that accommodates for measurements in a broad spectral region from 1000 cm$^{-1}$ to 4000 cm$^{-1}$. The spectra were recorded under the polarization combinations SSP (S polarized SF, S polarized visible, and P polarized IR beams), SPS, and PPP, and normalized by the non-resonant SF response from a gold substrate. The spectra presented in n.u. units have been normalized to the intensity of the $CD_3$ symmetric stretch in the SSP spectra, and can be directly compared.

**Surface hydronium concentration from the Gouy–Chapman theory**. The proton concentration at the surface $[H^+]_0$ depends on that in the bulk $[H^+]_\infty$, as well as the surface potential $\psi_0$,

$$[H^+]_0 = [H^+]_\infty \exp\left(\frac{-e\,\psi_0}{kT}\right) \tag{1}$$

where $e$, $k$, and $T$ are the elemental charge, the Boltzmann factor, and temperature, respectively. The surface potential is in turn related to the degree of dissociation a of the carboxylic acid monolayer ($\alpha$), and equilibrium constant $K_a$ for the reaction $HA \leftrightarrow A^- + H^+$,

$$K_a = \frac{[A^-]_0[H^+]_0}{[HA]_0} = \frac{\alpha}{1-\alpha}[H^+]_\infty \exp\left(\frac{-e\,\psi_0}{kT}\right)$$

$$\rightarrow \psi_0 = \frac{kT}{e}\left[\ln[H^+]_\infty - \ln K_a - \ln\left(\frac{1-\alpha}{\alpha}\right)\right] \tag{2}$$

The surface potential $\psi_0$ also depends on the concentration of ions in solution $[NaCl]_\infty$, and the surface charge density $\sigma$ (i.e., the degree of deprotonation), through the Grahame equation[52]:

$$\sigma = \frac{\alpha e}{A_M} = \sqrt{8\varepsilon\varepsilon_0 kT}\,\sin h\left(\frac{e\,\psi_0}{2kT}\right)\sqrt{[NaCl]_\infty}$$

$$\rightarrow \psi_0 = \frac{2kT}{e}\,\text{arc}\sin h\left(\frac{\alpha e/A_M}{\sqrt{[NaCl]_\infty\,8\varepsilon\varepsilon_0 kT}}\right) \tag{3}$$

where $\varepsilon$ and $\varepsilon_0$ are the dielectric constants of water and vacuum, respectively, and $A_M$ is the area per molecule (~20 Å$^2$ per molecule). The surface potential is determined by simultaneously solving Eq. (2) and Eq. (3) and setting the $pK_a$ to 5[26].

## Data availability

The datasets generated during and/or analyzed during this study are available from the corresponding author on reasonable request. The specific data used for plotting the Figures in the article can also be found in a Supplementary Dataset.

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

## Acknowledgements

This work is financially supported by the Swedish Foundation for Strategic Research (SSF) through the program "Future Research Leaders-5", and the Swedish Research Council (VR). We thank Ali Hassanali for extended and insightful discussions, and Robert Corkery and Istvan Furo for constructive feedback on the manuscript text.

## Author contributions

E.T. designed research. E.T, S.S. and A.S. performed research and analyzed data. E.T. wrote the article.

## Competing interests

The authors declare no competing interests.
