## [Peer Review File · Nature Communications]

Reviewers' comments:

Reviewer #1 (Remarks to the Author):

Manuscript ID: NCOMMS-19-2262408-T

Title: Identifying Eigen-like hydrated protons at negatively charged interfaces

Authors: Eric Tyrode, Sanghamitra Sengupta, and Adrien Sthoer

This paper reports on the discovery of Eigen-like hydrated protons at monolayer/water interfaces. The authors applied SFG spectroscopy to a dAA monolayer/water interface at ambient pH, and observed a clear signature of the hydrated proton at 2540 cm^{-1} especially in the SPS polarization combination. They also examined SES instead of dAA and obtained the same result. They assigned the 2540 cm^{-1} band to the antisymmetric stretch of hydrated H_3O^+ .

The reviewer is impressed by their thorough measurements in a wide wavenumber range with three different polarization combinations. (Actually many researchers carry out SFG spectroscopy in a limited wavenumber range only with SSP.) The present nice discovery can be attributed to their sincere attitude towards spectroscopy.

Nevertheless he has some concerns as follows:

Major

(1) The reviewer does not understand why the symmetric stretch of hydrated H_3O^+ does not appear in the SSP spectrum. He believes that the authors assigned the 2540 cm^{-1} band to the antisymmetric stretch because it appears in the SPS spectrum. Then the symmetric stretch has to be very visible in the SSP spectrum.

(2) It is also very strange that the OH stretch band around 3500 cm^{-1} is stronger in SPS than in SSP. As shown in their paper [25], for instance, the 3500 cm^{-1} OH band is much weaker in SPS than in SSP, which generally holds with no exception as far as he knows.

(3) Because of (1) and (2), he hesitates to completely agree to their assignment. He may worry about contamination from the organic compounds that they used to prepare the monolayers. More effort may be necessary to negate the possibility of the contamination (e.g. FTIR of each chemical).

Minor

(4) He does not agree to their using "asymmetric" that actually means lacking symmetry. They assigned the 2540 cm⁻¹ band to the E irreducible representation in the C_{3v} point group. Probably they wanted to mean non-totally symmetric (i.e. non-A₁) rather than lacking symmetry by using "asymmetric". No vibration lacks symmetry in the sense that any vibration belongs to an irreducible representation. Although many researchers use "asymmetric" in the same context as the authors, he thinks that it is more appropriate to use "antisymmetric" because the character of the E representation for the C₃ rotation is -1.

(5) Page 6, Line 8: redshift, Line 11: red-shifted

Shift from what?

(6) Page 7, Line 5: it is not negligible as predicted by the theoretical analysis

The theoretical analysis in Fig. 3b shows that it is negligible.

(7) Page 9, Line 11: the fact that it is more strongly observed

More than what?

(8) Page 11

Many typos in Methods, Solution preparation paragraph.

(9) Page 11, last sentence

Incomplete.

Reviewer #2 (Remarks to the Author):

This is largely a follow-up paper from an earlier report from this group on the sum frequency vibrational spectra of fatty acids at the air-water interface. It appears that this report focuses on a region near 2700 cm⁻¹ that was conspicuously absent in the first paper (and others I could find out

there). It now appears that at low NaCl concentration in the sub-phase, a strong band near 2700 cm^{-1} appears that is assigned to the "Eigen" form of a hydrated proton based on the behavior of gas phase clusters. This is a very interesting observation, as it seems that this region has been ignored in the many surface spectroscopic investigations of the surfactant interface involving acids in (and on) water. It is thus important to clarify what is happening in this range, and I suspect that the present paper will be of interest to a wide community looking at these effects. The fact that the long chain sulfate system displays essentially the same band indeed supports the author's assertion that the band is not due to intimate contact with the acid scaffolds. I have a few comments, however that should be addressed prior to publication:

1. Where is the neutral acid OH stretch expected to occur? It would seem that, if most of the acid is undissociated, there should be an absorption, possibly quite red-shifted from the water OH envelope.
2. Is there a possibility that the deprotonated acid head group binds to a nearby intact acid to form a proton-bound dimer? It would seem that in the monolayer regime, such effects would be dominant speciation of the conjugate base.
3. The authors correctly point out in larger gas phase water clusters, the Eigen bands red-shift by about 600 cm^{-1} . Given that, why would a bulk interfacial band be expected to occur at the same position as in the isolated Eigen? Are there any calculations that support this assignment?
- 4 On page 4 line 13, "the monolayer is less than 0.5% deprotonated and the carboxylate modes are below the detection limit", could you clarify if the absence of the carboxylate modes was used as evidence to reach the conclusion that less than 0.5% deprotonation occurred or the 0.5% value was calculated?
- 5 On page 8 line 18, "...with the three hydrogens directed towards the negatively charged interface" Could you go into a little more detail on how this conclusion was drawn?
- 6 Page 9 Fig 5. It might be worth it to point out that the peak areas in panel b seems to agree with the predicted concentration in panel a.

There are also a few typos:

1. On page 2 line 16, insert "are" in "linked to Zundel and Eigen structures, and are also used"

Pag2 line 10: ...same time makes...

Page 2: Ultracold is too strong. Just "cryogenically cooled to 20K" would be better

Page 7: an intrinsically ..

Page 9: ...subphase indicates...

Reviewer #3 (Remarks to the Author):

This manuscript beautifully presents evidence of an Eigen resonance for both H₂O and D₂O systems with surfactant monolayers. The data is highly compelling to say the least. The rigor for which the authors present the scenarios of interfacial protonation and comparison to literature furthers their arguments. The SFG spectra are particularly clean and provide indisputable evidence of new resonances that appear to be consistent with the author's assertion of Eigen like interfacial assignment.

*There is an important correction in that the authors' incorrectly state on page 3 that the proton continuum region less than 3000 has been neglected. In fact it was published by Allen and coworkers in reference 20 (JPC Lett, 2007, 111, 8814-8826 Levering et al.; See Figure 8, 2400-3200 resonance of the acids is shown). This should be discussed in addition to being referenced adequately in light of the assertions made in the paper.

* Also, on page 8 at the bottom of the page the authors discuss that "at a negatively charged surface, the concentration is several orders of magnitude higher". This statement is confusing as the main chemical system is the acid form to begin with; the negative surface is being formed according to its apparent pK_a, as discussed with using the Gouy Chapman model in the next sentences. Is this statement on page 8 really correct?

* On a minor note, I suggest adding some detail of the frequency of the assigned Eigen for both D₂O and H₂O systems in the abstract.

Reviewer #4 (Remarks to the Author):

The manuscript by Tyrode's group on the Eigen-like structure of the hydrated proton at negatively charged vapour/water interfaces reports static $|\chi_2|^2$ Sum Frequency Generation experiments of negatively charged water/vapour interfaces made of Langmuir monolayers exposing carboxylic or sulphate groups to the solution sub-phase. The claim of the work is that the experimentally observed SFG band at roughly 2540 cm^{-1} arises from eigen-like protons located at the interface.

Though the experiments are well-conducted and the authors make efforts to convince the readers that this signature arises from protons at the interface and not from the Langmuir monolayers, I do not see this work as being convincing on the eigen-like structure of the protons, neither convincing on the fact that the protons are indeed located within the Stern layer rather than solvated within the diffuse layer (or bulk-like oriented Diffuse Layer from Tian/Shen's PRL 2016 paper denomination), or resulting in a mixture of both locations. I find this work incremental in the general search of demonstrating that protons are located at the air/water interface (and I would even provocatively say that there are now many convincing experimental and theoretical works assessing this conclusion), without the decisive arguments that would close a never-ending debate. Therefore, I do not believe that this paper is fitted for the broad audience of Nature Communications, but rather it should be published in a more specialized journal where this specialized debate is of more interest.

Below are some further remarks on the paper.

Tyrode's group is conducting static 'non-phase resolved' SFG experiments where not only the real and imaginary parts of the χ_2 susceptibility are mixed but also without any direct information on the orientation of the O-H oscillators contributing to the signals. They also do not take into account the decisive work from Tian/Shen (PRL 2016) and subsequent works in the literature that insist on the need for decomposing the SFG signatures into the immediate top-most interface (called Stern layer, Helmholtz layer, Binding Interfacial Layer, depending on the authors) and the subsequent bulk-like oriented Diffuse Layer (taking the denomination of Tian/Shen) in order to interpret the only relevant spectral signatures arising from the top-most interface. In the present work, the authors do tell us that the very low ionic concentration used in the experiments would lead to interferences that would cancel the Diffuse Layer contribution to the SFG signal, but without a clear demonstration. I therefore find no clear evidence in this paper that the 2640 cm^{-1} signature is definitely arising from the top-most layer and not from the subsequent Diffuse Layer or a mixture of both contributions.

In the introduction of the paper where the authors convince the readers that probing the 3000-4000 cm^{-1} stretch O-H signatures have not given sufficient conclusive arguments on the location of the proton at the interface, thus the need for probing lower frequency signatures, the authors do not cite theory papers from e.g. Hassalani's group or Gaigeot's group both pointing to a special organization of the water at the interface with the air that they both rather convincingly relate to the fact that there are indeed no signatures that can be recorded in the O-H stretch domain that would point to a direct signature of protons at the interface. I encourage the authors to cite these works.

Furthermore, these authors as well as Paesani's and Kuhne's theory works point to rather fast and easy proton transfers at the air/water interface. In these conditions, I must confess that I always find the debate on the eigen-/zundel-like structures of the

protons (whether in liquid or at interfaces) rather futile as the protons are highly dynamical at room temperature and therefore the static 'reduced' structure picture seems rather pointless, and hence the assignment of the 2640 cm⁻¹ signature to one given (here eigen-like) structure seems dangerous. It is all the more puzzling as the (cold) clusters experiments by Johnson's group cited by the authors show much lower frequencies for the eigen-like proton as soon as the number of water in the clusters increase, as would be the case at the interface by just adding up the influence of the immediate water neighbours to the solvated proton. So in a nutshell, why is it that the SFG has a rather sharp 2640 cm⁻¹ band, pointing to a rather static picture of the protons at the interface while a more broader/delocalized band could be expected for dynamical protons? Could it be that the choice of the investigated charged monolayers induce such biased static picture, not of enough general conclusion?

Journal: Nature Communications. Manuscript ID : **NCOMMS-19-2262408-T**
Title: "Identifying Eigen-like hydrated protons at negatively charged interfaces"
Author(s): Tyrode, Eric; Sengupta, Sanghamitra; Stoeber, Adrien

We would like to thank the reviewers for carefully reading our manuscript and for providing helpful comments and suggestions. We have revised the manuscript on the basis of their comments and provide point-by-point replies below. The original comments of the referees are in blue and our responses are black fonts.

Reviewer 1.

This paper reports on the discovery of Eigen-like hydrated protons at monolayer/water interfaces. The authors applied SFG spectroscopy to a dAA monolayer/water interface at ambient pH, and observed a clear signature of the hydrated proton at 2540 cm^{-1} especially in the SPS polarization combination. They also examined SES instead of dAA and obtained the same result. They assigned the 2540 cm^{-1} band to the antisymmetric stretch of hydrated H_3O^+ . The reviewer is impressed by their thorough measurements in a wide wavenumber range with three different polarization combinations. (Actually, many researchers carry out SFG spectroscopy in a limited wavenumber range only with SSP.) The present nice discovery can be attributed to their sincere attitude towards spectroscopy.

We thank the reviewer for carefully reading the manuscript and his positive general assessment of our work.

Nevertheless, he has some concerns as follows:

Major

(1) The reviewer does not understand why the symmetric stretch of hydrated H_3O^+ does not appear in the SSP spectrum. He believes that the authors assigned the 2540 cm^{-1} band to the antisymmetric stretch because it appears in the SPS spectrum. Then the symmetric stretch has to be very visible in the SSP spectrum.

We appreciate the reviewer's comment as he raises a point that also concerned us since the moment we first observed the band. The assignment to an antisymmetric stretch is based on two independent sources of information. The first relies on experiments and simulations on gas phase clusters, where the equivalent band is assigned to a doubly degenerate antisymmetric OH stretch of the Eigen core (see for instance references #1, #3, #14, #34, and #39 in manuscript). Interestingly, it is worth noting that the accompanying symmetric stretch is not experimentally resolved in any of the cluster studies, with theory suggesting it is slightly red-shifted (ref#3), overlaps (ref#39), or even slightly blue-shifted (ref#1) relative to the antisymmetric stretch. Moreover, simulations at the harmonic and double-harmonic level indicate that the IR cross-section for the symmetric stretch is almost two orders of magnitude lower than for the antisymmetric mode (~3%, ref#9 and ref#40). Given that the SF intensity has a square dependence on the IR transition moment, the intensity of the symmetric mode is then expected to be considerably weaker. Having said this, the SF intensity also depends on the Raman polarizability tensor elements, which are expected to be higher for the symmetric stretch. However, due to the non-Condon effect, Raman cross-sections, relative to IR, substantially decrease with a lowering of the OH frequency of vibration, as it is indeed the case for

hydronium stretches (Ref#4). Consequently, both elements (i.e. IR cross-section, and non-Condon effect) indicate that the SF cross-section for the H_3O^+ symmetric stretch will be significantly lower than for the antisymmetric mode.

The second point we use to substantiate the assignment is the SF spectra collected at different polarization combinations, as highlighted by the reviewer. The fact that the 2540 cm^{-1} band is more strongly observed in the SPS polarization combination is compatible with assigning the band to the antisymmetric stretch of the H_3O^+ Eigen core (C_{3v} symmetry. Figure 3 in manuscript). The same orientation analysis indicates that the antisymmetric stretch should not be observed in the SSP polarization combination. However, since a weak band is resolved in the latter polarization, we suggest it originates from the symmetric stretch of H_3O^+ , which should appear, as mentioned by the reviewer, in the SSP polarized spectra. In the original text of the manuscript and SI these elements were mentioned when discussing the assignment. However, following the reviewer's comment, we extend this description and make it much more explicit in the revised version of the manuscript.

The concerned text in Page 7, now reads:

“However, though the experimental intensity in SSP is significantly weaker than for SPS (Figure 3a), it is not negligible as predicted by the theoretical analysis for low tilt angles (Figure 3b). The discrepancy can be accommodated by considering potential contributions from the symmetric stretch of the C_{3v} Eigen core, which when aligned close to the surface normal, should be preferentially observed in the SSP polarized spectra (see SI for details).^{35, 37} Although the symmetric stretch has not been experimentally resolved in any of the previous IR cluster studies, it has been theoretically estimated to lie just $\sim 38\text{ cm}^{-1}$ red-shifted,³ overlapping,³⁸ or even blue-shifted¹ relative to the antisymmetric mode, with a cross-section that is almost two orders of magnitude lower than for the degenerate antisymmetric stretch.^{9, 39} Given that the SF intensity has a square dependence on the IR and Raman transition moments,^{35, 36} the significantly lower IR cross-section will have an obvious effect on the intensity of the symmetric stretching mode. Moreover, even though the mode is expected to be stronger in Raman, due to the non-Condon effect,^{4, 40} the contribution of the IR component relative to that of Raman in the SF cross-section, becomes more important at the lower OH stretching frequencies of the Eigen core. Consequently, both elements (i.e. low IR cross-section, and non-Condon effect), indicate that the SF cross-section of the H_3O^+ symmetric stretch will be significantly lower than for the antisymmetric mode.”

(2) It is also very strange that the OH stretch band around 3500 cm^{-1} is stronger in SPS than in SSP. As shown in their paper [25], for instance, the 3500 cm^{-1} OH band is much weaker in SPS than in SSP, which generally holds with no exception as far as he knows.

Although in the case of pure water, the SPS signal is indeed much weaker than SSP (reference #25, as correctly mentioned by the reviewer), this is not always the case. For instance, previously reported SF spectra on surfactant solutions (JACS, 2005, 127, 16848) show that the intensity of the SPS polarized spectra can be higher than that for SSP, particularly at high frequencies ($>3450\text{ cm}^{-1}$). Having said this, following the reviewer's comment, we carefully re-examined the spectra presented in Figure 1a, and noticed that the relative intensity of the water OH band at $\sim 3500\text{ cm}^{-1}$ in the SPS spectrum, was higher than the average of multiple repeat experiments. We trace the problem to an improper stitching of our spectra. The SF data presented in Figure 1, results from combining three spectra collected at different grating

positions. Although we make sure there is sufficient overlap between the different spectra, the stitching procedure was incorrectly done in the SPS data we presented in the submitted manuscript. A graphical description of this oversight is presented in Figure R1 below. Figure R1 left, shows how the data was incorrectly stitched in the submitted version (Figure 1a in manuscript). Figure R1 right shows how it is (correctly) done in the revised version, where the relative intensity of the OH water band at 3500 cm^{-1} has now decreased by approximately 40%. In this respect, we are very grateful to the reviewer for having brought up this issue. We note, however, that this unfortunate oversight has no consequences in any of the following sections and conclusions in our manuscript.

Figure R1. The figure in the left shows how the spectra was incorrectly stitched in the data presented in the original manuscript. The figure on the right shows how it is done in the revised version.

Figure R2. Left: Figure 1a presented in the original manuscript. Right: Revised Figure 1a after correcting the spectral stitching.

(3) Because of (1) and (2), he hesitates to completely agree to their assignment. He may worry about contamination from the organic compounds that they used to prepare the monolayers. More effort may be necessary to negate the possibility of the contamination (e.g. FTIR of each chemical).

Having addressed points (1) and (2) above, we hope to have dissipated the reviewer's doubts. Regarding the potential effect of contaminants, we must stress that we have considered this possibility very carefully. For instance, we have used chemicals with the highest purity available and made certain that organic contaminants are not present in the salts (i.e. all salts were baked at 500 oC before use. See additional considerations in reference #27 from previous studies in our group). We have observed the Eigen band with different fatty acid batches, either per-deuterated, or per-protonated, as well as with a fatty sulfate (see Figure 4). This discards any possibility of random contamination in the chemicals used. Moreover, the amount of material added to create the Langmuir monolayers is minute (i.e. single molecular thick layer), which further reduces the already negligible probabilities of finding a persistent contaminant in the bulk solution.

Minor

(4) He does not agree to their using "asymmetric" that actually means lacking symmetry. They assigned the 2540 cm^{-1} band to the E irreducible representation in the C_{3v} point group. Probably they wanted to mean non-totally symmetric (i.e. non-A1) rather than lacking symmetry by using "asymmetric". No vibration lacks symmetry in the sense that any vibration belongs to an irreducible representation. Although many researchers use "asymmetric" in the same context as the authors, he thinks that it is more appropriate to use "antisymmetric" because the character of the E representation for the C_3 rotation is -1 .

We agree with the reviewer that it is more appropriate to use "antisymmetric" instead of "asymmetric". The term was corrected throughout the manuscript.

(5) Page 6, Line 8: redshift, Line 11: red-shifted. Shift from what?

The concerned sentence was rewritten following one of the comments from reviewer #2.

(6) Page 7, Line 5: it is not negligible as predicted by the theoretical analysis
The theoretical analysis in Fig. 3b shows that it is negligible.

The sentence was rewritten following comment #1 above.

(7) Page 9, Line 11: the fact that it is more strongly observed. More than what?

This sentence was rewritten following a comment from reviewer#4.

(8) Page 11. Many typos in Methods, Solution preparation paragraph.

The section was revised and typos corrected.

(9) Page 11, last sentence Incomplete.

The incomplete sentence was removed. We thank the reviewer for the overall insightful and constructive feedback.

Reviewer 2.

This is largely a follow-up paper from an earlier report from this group on the sum frequency vibrational spectra of fatty acids at the air-water interface. It appears that this report focuses on a region near 2700 cm^{-1} that was conspicuously absent in the first paper (and others I could find out there). It now appears that at low NaCl concentration in the sub-phase, a strong band near 2700 cm^{-1} appears that is assigned to the “Eigen” form of a hydrated proton based on the behavior of gas phase clusters. This is a very interesting observation, as it seems that this region has been ignored in the many surface spectroscopic investigations of the surfactant interface involving acids in (and on) water. It is thus important to clarify what is happening in this range, and I suspect that the present paper will be of interest to a wide community looking at these effects. The fact that the long chain sulfate system displays essentially the same band indeed supports the author’s assertion that the band is not due to intimate contact with the acid scaffolds.

We appreciate the reviewer’s assessment that the subject matter of our paper is important, and of potential interest to a wide community.

I have a few comments, however that should be addressed prior to publication:

1. Where is the neutral acid OH stretch expected to occur? It would seem that, if most of the acid is undissociated, there should be an absorption, possibly quite red-shifted from the water OH envelope.

The OH stretch of the carboxylic acid headgroup when hydrogen bonded, is expected to give rise to a weak broad band centred between 2900 and 2950 cm^{-1} . The band is strong in IR but barely detectable in Raman, making it very weak in SF. See for instance JPCB 2005, 109, 321 or Chem. Phys. Letters 1998, 286, 1. In our fatty monolayers measured at pH ~6 the band cannot be readily resolved in our spectra, although it could be probably linked to the weak broad band centred at ~2850 cm^{-1} . We note that if the OH is not hydrogen bonded the peak sharpens and shifts to 3580 cm^{-1} (JPCB 2005, 109, 321). To take into account the reviewer’s comment we have added a sentence on page 4, following the description of the spectra shown in Figure 1. The added sentence reads:

“(the OH stretch from the hydrogen bonded carboxylic acid gives rise to a weak broad band centred at ~2900-2950 cm^{-1}).³⁰”

2. Is there a possibility that the deprotonated acid head group binds to a nearby intact acid to form a proton-bound dimer? It would seem that in the monolayer regime, such effects would be dominant speciation of the conjugate base.

Proton-bound dimers are more commonly observed in the absence of water and in systems where the carboxylate headgroups can relatively freely orient with respect to each other (i.e. ionic liquids or gas clusters. See for instance Chem. Commun. 2011, 47, 3222–3224., Int. J. Mass Spectrom. 2011, 300, 91– 98 ; or Angew. Chem. Int. Ed. 2018, 57, 10615 –10619). In our monolayers the headgroup is in contact with water, and the twenty carbon long chain offers limited, if any flexibility for the headgroups to form, for example, the cyclic dimer structures commonly observed with shorter chain carboxylic acids (see for instance ref #38 or proton-bound formate cluster from Angew. Chem. Int. Ed. 2018, 57, 10615 –10619). We have previously shown (ref#27), however, that upon deprotonation the average orientation of the uncharged carboxylic acid moieties slightly varies, an effect that we attributed to interaction

between the C-OH groups with neighboring charged carboxylates. Nonetheless, the OH is not “bound” to the carboxylate (this would have consequences in several spectral features, as described for example in the cluster studied above: *Int. J. Mass Spectrom.* 2011, 300, 91– 98 ; or *Angew. Chem. Int. Ed.* 2018, 57, 10615 –10619). We are confident that such type of interactions are not related to the band assigned to the Eigen species. Accordingly, no changes were made in the manuscript.

3. The authors correctly point out in larger gas phase water clusters, the Eigen bands red-shift by about 600 cm^{-1} . Given that, why would a bulk interfacial band be expected to occur at the same position as in the isolated Eigen? Are there any calculations that support this assignment?

The reviewer raises an important point. Indeed, the isolated $\text{H}^+(\text{H}_2\text{O})_n$ $n=4$ Eigen cation cluster shows the well-resolved antisymmetric stretch at 2660 cm^{-1} . However, the same band, linked to the Eigen motif, is also clearly observed in clusters with $n=9$, $n=10$ and $n=11$, and remains visible in the spectra of larger clusters, such as $n=12$ and $n=24$ (Ref #1, #10, #34). The presence of the 2660 cm^{-1} band appears to be highly dependent on the symmetry of the solvation shell hydrating the hydronium core. For instance, for a gas cluster with $n=5$, the asymmetric solvation of the core causes the antisymmetric stretch to split with one peak found at 2750 cm^{-1} and the other below 1600 cm^{-1} (ref#34). From $n=6$ to $n=8$ the structure is more Zundel-like, and the characteristic vibrational features are even further shifted. For cluster with $n>10$ up to $n=28$ (apparently the largest cluster before bulk features become dominant), a so-called “surface Eigen clathrate cage” is formed, and the OH vibrational signatures of the H_3O^+ often occur between two general regions, around 2000 cm^{-1} and 2600 cm^{-1} (ref#34). In the original manuscript this effect was summarized in a sentence that was not sufficiently explicit, and easily misinterpreted. The concerned sentence on page 6 previously read: “*This band is observed to redshift up to $\sim 600\text{ cm}^{-1}$ when increasing the number of water molecules in clusters that display an Eigen-like motif.*”³⁴” The sentence has now been edited as follows:

“This band is also observed in larger gas clusters displaying an Eigen-like motif where the hydronium core is predominantly symmetrically solvated (i.e. $\text{H}^+(\text{H}_2\text{O})_{n=9-12}$), and remains detectable in even bigger clusters (i.e. $n=24$), though contributions red-shifted by up to $\sim 600\text{ cm}^{-1}$ also become apparent in the spectra.”^{1,34}

Regarding the questions asked by the reviewer, we note that the peak position of the band assigned to the antisymmetric Eigen stretch in our study is $\sim 120\text{ cm}^{-1}$ red-shifted from that observed in the gas phase, so they do not appear at the same position. However, in terms of calculations we rely on previous extensive and careful studies on gas phase clusters and to some extent, those recently performed in concentrated bulk acid solutions where theory was compared with experimental IR-MCR and Raman-MCR spectra (ref#4). In the latter study, a broad band ($\pm 500\text{ cm}^{-1}$) centered at 2500 cm^{-1} was linked to Eigen-like species. We had not explicitly made this link in the manuscript, so we have added a sentence mentioning this point as additional support to our assignment on page 9, which reads:

*“The centre position is also in agreement to that proposed from calculated IR and Raman spectra for structures that can be classified as Eigen-like.”*⁴

Finally, we would like to note that the theoretical modelling of a system similar to that measured with VSFS (i.e. carboxylic acid moiety at pH 6 and $1\mu\text{M NaCl}$), is currently not feasible, at least not with a sufficient high-level of theory (for instance AIMD). The number of water molecules required in a simulation box to reach 1 uM concentration of H^+ is prohibitive, and if

concentrations are increased beyond 10 μM ($\text{pH} < 5$), under equilibrium conditions, the proton should not be found in its free form but condense on the carboxylate moiety ($\text{pK}_a = 5$). Moreover, current force fields available for modelling carboxylate interactions with monovalent metal cations (let aside H^+) tend to overestimate the interactions (see for instance ref#27). We hope the results presented in this work will trigger an interest for additional theoretical and simulations studies on hydrated protons close to charged interfaces.

4 On page 4 line 13, “the monolayer is less than 0.5% deprotonated and the carboxylate modes are below the detection limit”, could you clarify if the absence of the carboxylate modes was used as evidence to reach the conclusion that less than 0.5% deprotonation occurred or the 0.5% value was calculated?

The degree of deprotonation was calculated using the Gouy-Chapman theory, which was found to accurately model the degree of deprotonation up to a concentration of 50 mM NaCl, as well as its pH dependence. The degree of deprotonation can be experimentally measured from approximately 1%, which corresponds to 0.1 mM NaCl. This is extensively discussed in ref#26, but it is not explicitly mentioned in the sentence highlighted by the reviewer. This has been clarified in the text. The concerned sentence on page 4 now reads:

“At pH ~ 6 and a NaCl concentration of 1 μM , the carboxylate modes are below the detection limit, yet the monolayer is expected to be $\sim 0.5\%$ deprotonated, as estimated from the Gouy-Chapman model.²⁶”

5 On page 8 line 18, “...with the three hydrogens directed towards the negatively charged interface” Could you go into a little more detail on how this conclusion was drawn?

The sentence was rewritten as follows:

“Although the net polar orientation can not be directly obtained from the homodyne SF spectra presented,³⁹ the three hydrogens will most certainly be directed towards the negatively charged interface.”

A sketch of the hydronium ion orientation inferred from the orientation analysis was also included in Figure 3b. The revised figure is reproduced here:

6 Page 9 Fig 5. It might be worth it to point out that the peak areas in panel b seem to agree with the predicted concentration in panel a.

The following sentence was added to the legend of Figure 5.

“Note that the H_3O^+ antisymmetric stretch intensity follows the same trend predicted for the surface proton concentration as a function of salt.”

There are also a few typos:

1. On page 2 line 16, insert “are” in “linked to Zundel and Eigen structures, and are also used”

Page 2 line 10: ..same time makes...

Page 2: Ultracold is too strong. Just “cryogenically cooled to 20K” would be better

Page 7: an intrinsically ..

Page 9: ...subphase indicates...

We thank the reviewer for highlighting the various typos, which were corrected in the revised version of the manuscript.

Reviewer 3.

This manuscript beautifully presents evidence of an Eigen resonance for both H₂O and D₂O systems with surfactant monolayers. The data is highly compelling to say the least. The rigor for which the authors present the scenarios of interfacial protonation and comparison to literature furthers their arguments. The SFG spectra are particularly clean and provide indisputable evidence of new resonances that appear to be consistent with the author's assertion of Eigen like interfacial assignment.

We are grateful for the reviewer's positive assessment of our study. We also thank the reviewer for carefully reading the manuscript and providing constructive feedback.

*There is an important correction in that the authors' incorrectly state on page 3 that the proton continuum region less than 3000 has been neglected. In fact it was published by Allen and coworkers in reference 20 (JPC Lett, 2007, 111, 8814-8826 Levering et al.; See Figure 8, 2400-3200 resonance of the acids is shown). This should be discussed in addition to being referenced adequately in light of the assertions made in the paper.

We thank the reviewer for directing us to the data presented in Ref#20, and apologize for this oversight. We have edited a sentence at the end of the introduction in Page 2 that now reads:

“With the exception of a couple of studies on neat acid solutions,^{20, 25} where no resolved resonant features were detected, the “proton continuum” region ($<3000\text{ cm}^{-1}$) on surfaces has largely been neglected.”

An additional sentence related to ref#20 was also added on support to our arguments on page 8:

“For instance, the VSF spectrum of pure water, as well as that of solutions of HCl at pH 2, and HBr at pH -0.3, show no resolved resonant features in the proton continuum range,^{20, 25} highlighting the importance of having a (partly) charged Langmuir monolayer for detecting the hydrated proton at the surface.”

* Also, on page 8 at the bottom of the page the authors discuss that "at a negatively charged

surface, the concentration is several orders of magnitude higher". This statement is confusing as the main chemical system is the acid form, to begin with; the negative surface is being formed according to its apparent pKa, as discussed with using the Gouy Chapman model in the next sentences. Is this statement on page 8 really correct?

We are confident that the statement is correct, as the apparent pKa of the fatty acid monolayer is higher than the intrinsic pKa, due to the fact that the proton concentration at the surface is higher than in the bulk. Following the reviewer's comment, we have slightly edited the sentence to make it more explicit:

"At the negatively charged surface, however, the proton concentration is orders of magnitude higher."

* On a minor note, I suggest adding some detail of the frequency of the assigned Eigen for both D2O and H2O systems in the abstract.

The abstract was edited following the reviewer's suggestion. The added sentence reads:

"Centered at $\sim 2540\text{ cm}^{-1}$, the band is seen to shift to $\sim 1875\text{ cm}^{-1}$ when forming D_3O^+ upon isotopic substitution."

Reviewer 4.

The manuscript by Tyrode's group on the Eigen-like structure of the hydrated proton at negatively charged vapour/water interfaces reports static $|\chi|^2$ Sum Frequency Generation experiments of negatively charged water/vapour interfaces made of Langmuir monolayers exposing carboxylic or sulphate groups to the solution sub-phase.

The claim of the work is that the experimentally observed SFG band at roughly 2540 cm^{-1} arises from eigen-like protons located at the interface. Though the experiments are well-conducted and the authors make efforts to convince the readers that this signature arises from protons at the interface and not from the Langmuir monolayers, I do not see this work as being convincing on the eigen-like structure of the protons, neither convincing on the fact that the protons are indeed located within the Stern layer rather than solvated within the diffuse layer (or bulk-like oriented Diffuse Layer from Tian/Shen's PRL 2016 paper denomination), or resulting in a mixture of both locations. I find this work incremental in the general search of demonstrating that protons are located at the air/water interface (and I would even provocatively say that there are now many convincing experimental and theoretical works assessing this conclusion), without the decisive arguments that would close a never-ending debate. Therefore, I do not believe that this paper is fitted for the broad audience of Nature Communications, but rather it should be published in a more specialized journal where this specialized debate is of more interest.

We acknowledge the reviewer's opinion but respectfully disagree. We are convinced our findings are not incremental but instead an important breakthrough. One of the key aspects of our work is that we observe in the SF spectra a resolved and relatively narrow feature within the proton continuum region, that we assigned to a specific hydrated proton species. This is very different from what has been observed in the bulk or previous surface studies, where a broad, almost featureless continuum is seen instead. The "never-ending debate" the reviewer

refers to revolves around whether hydronium or hydroxide ions adsorb to the neat interface (see for instance the review in ref #10), a point that we do not address here. At the **negatively charged interfaces** we discuss in our manuscript, there is no doubt that the protons preferentially adsorb. The rather striking point is that they can be spectroscopically detected, particularly when their concentration in the bulk is just a few μM . Additionally, we show that the same band appears in surfaces exposing different functionalities, which not only supports our conclusions but also broadens the scope and relevance of the findings. Given the ubiquity of protons in solution and the novelties of our work, we strongly believe this manuscript is well suited for the broad audience of Nature Communications.

Below are some further remarks on the paper.

Tyrodé's group is conducting static 'non-phase resolved' SFG experiments where not only the real and imaginary parts of the χ^2 susceptibility are mixed but also without any direct information on the orientation of the O-H oscillators contributing to the signals.

The reviewer implies with her/his comment that homodyne SF experiments, like the ones we carried out in this work, do not provide sufficient information to substantiate our conclusions. We respectfully disagree. Not only does the vast majority of SF experiments reported in the literature use homodyne SF, but carrying out heterodyne SF studies in the broad spectral regions reported in this manuscript is simply not technically possible at the moment (i.e. reference phase stability). Moreover, as the relatively narrow band we assigned to the Eigen proton is substantially red-shifted from the broad OH water stretches, the main additional information heterodyne SF would provide is the net polar orientation (i.e. hydrogens pointing up or down). This was already stated on page 8. Having said this, performing VSFS measurements in its heterodyne, scattering, and particularly time-resolved configurations could be useful, incremental, next steps, as summarized in our concluding paragraph on page 10.

The reviewer also states that no direct orientation information can be obtained. We disagree. The theory is well established, and the number of published papers using homodyne SF to determine the orientation of specific functional groups can be counted in the hundreds. This is actually exemplified in Figure 3 in the manuscript and extended in the SI. What homodyne SF does not directly provide is the absolute polar orientation as mentioned in the previous paragraph.

They also do not take into account the decisive work from Tian/Shen (PRL 2016) and subsequent works in the literature that insist on the need for decomposing the SFG signatures into the immediate top-most interface (called Stern layer, Helmholtz layer, Binding Interfacial Layer, depending on the authors) and the subsequent bulk-like oriented Diffuse Layer (taking the denomination of Tian/Shen) in order to interpret the only relevant spectral signatures arising from the top-most interface.

We respectfully disagree. We do cite the excellent work by Wen et al. (REF#44, which in the previous submission was REF#40) as well as that from Gonella/Roke (REF#32, published independently at very much the same time), in the context of potential contributions from molecules in the immediate surface and within the diffuse double layer (see Page 10 first paragraph).

In the present work, the authors do tell us that the very low ionic concentration used in the experiments would lead to interferences that would cancel the Diffuse Layer contribution to the SFG signal, but without a clear demonstration. I therefore find no clear evidence in this paper

that the 2640 cm⁻¹ signature is definitely arising from the top-most layer and not from the subsequent Diffuse Layer or a mixture of both contributions.

The theoretical demonstration the reviewer requests is already found in REF#32, as well as in REF#33, which extends the results of REF#32 for conditions where the Debye-Hückel approximation is not valid, as is the case of our monolayers. The conclusions from REF#32 (and REF#33) have been experimentally confirmed in several recent studies, such as for instance REF#26, REF#33, and REF#45. At the lowest ionic strengths at which experiments were carried out, the signal from molecules within the diffuse double layer should mostly cancel, and progressively contribute more as the ionic strength is increased (at least for concentrations below 1 mM). However, as shown in Figure 5b, the intensity of the Eigen band decreases with ionic strength, suggesting that the signal does not originate from the diffuse double layer. Considering the reviewer's comment we added an additional sentence to the concerned section to make this point more explicit. Page 10 top paragraph (the new elements are underlined).

“However, the fact that the Eigen band is more strongly observed on a 1 μM NaCl subphase than at higher salt concentrations (Figure 5b), suggests that the origin is from close proximity to the interface. Had the signal been generated from within the diffuse double layer, due to destructive interference, it would have increased with ionic strength in the concentration range considered as experimentally confirmed elsewhere^{26, 42}

In the introduction of the paper where the authors convince the readers that probing the 3000-4000 cm⁻¹ stretch O-H signatures have not given sufficient conclusive arguments on the location of the proton at the interface, thus the need for probing lower frequency signatures, the authors do not cite theory papers from e.g. Hassalani's group or Gaigeot's group both pointing to a special organization of the water at the interface with the air that they both rather convincingly relate to the fact that there are indeed no signatures that can be recorded in the O-H stretch domain that would point to a direct signature of protons at the interface. I encourage the authors to cite these works.

The authors believe that papers related to the neat surface of concentrated acid solutions, as those mentioned by the reviewer, but which include many others (both experimental and theoretical), are not directly relevant for the current study on negatively charged interfaces. However, given that SF spectroscopy is the main technique used in this work, we believe important, for comparison purposes, to mention previous SF studies that have directly or indirectly targeted the proton. Having said this, we note that we already cite two of A. Hassalali's papers: REF#4 and REF#10, with the latter being a recent review on the current understanding.

Furthermore, these authors as well as Paesani's and Kuhne's theory works point to rather fast and easy proton transfers at the air/water interface. In these conditions, I must confess that I always find the debate on the eigen-/zundel-like structures of the protons (whether in liquid or at interfaces) rather futile as the protons are highly dynamical at room temperature and therefore the static 'reduced' structure picture seems rather pointless, and hence the assignment of the 2640 cm⁻¹ signature to one given (here eigen-like) structure seems dangerous. It is all the more puzzling as the (cold) clusters experiments by Johnson's group cited by the authors show much lower frequencies for the eigen-like proton as soon as the number of water in the clusters increase, as would be the case at the interface by just adding up the influence of the immediate water neighbours to the solvated proton. So in a nutshell, why is it that the SFG has a rather

sharp 2640 cm^{-1} band, pointing to a rather static picture of the protons at the interface while a more broader/delocalized band could be expected for dynamical protons? Could it be that the choice of the investigated charged monolayers induce such biased static picture, not of enough general conclusion?

First, we would like to stress that the SF signal does not originate from “static” molecules. We are collecting 100 femtosecond long snapshots of interfacial molecular configurations that have a preferred orientation. The fact that we observe a relatively narrow band does not imply that the Eigen species is static (i.e. long lived), it could be a very dynamic process. It all depends on the time scale, which in our case is in the subpicosecond range. This is true, not only for this study, but for most other (femto) SF measurements found in the literature.

Second, the nature of the interface is clearly of importance as we discussed throughout the manuscript, starting with the title. We observe the relatively narrow Eigen band at negatively charged interfaces, but not on the neat surface of pure acid solutions. We also demonstrate that it is not limited to a specific functional group, but it is observed at least on surfaces exposing either a carboxylic acid or a sulphate headgroups to solution. This already significantly broadens the implications, particularly when considering the importance of these moieties in biology (particularly the first one).

Third, as discussed in more detail in our response to question 3 of reviewer #2, in the gas clusters the band centered at approx. 2650 cm^{-1} is not only observed in the isolated Eigen ($n=4$), but also in larger clusters such as those with $n=9$, $n=10$, $n=11$, $n=12$, and even $n=24$ (several sentences were added in the text to make this point more explicit. See response to reviewer#2). Moreover, the band we assigned to the antisymmetric H_3O^+ stretch is centred at 2540 cm^{-1} (120 cm^{-1} red-shifted from that in the cluster studies).

Finally, regarding the question why would the protons preferentially adopt an Eigen-like configuration close to these negatively charged interfaces, we can only speculate at the moment, but hope that this work would trigger additional experimental and particularly theoretical (MD) interest on the properties of protons next to charged interfaces.

REVIEWERS' COMMENTS:

Reviewer #1 (Remarks to the Author):

The reviewer is satisfied with the reply.

Reviewer #2 (Remarks to the Author):

This is an interesting paper, and while I am not convinced that the $\text{H}_3\text{O}^+\text{OH}$ stretches will occur this high in energy above the surface-bound Eigen ion in clathrate water clusters, the air-water interface is indeed a distinct regime and I support publication of this paper so we can begin the discussion and get the ball rolling. It might be right!

Reviewer #3 (Remarks to the Author):

The authors have duly addressed all points that all reviewers raise. The manuscript data and interpretation are compelling and of interest to a broad community, and I recommend publication.